# Augmented-Reality-Based 3D Emotional Messenger for Dynamic User Communication with Smart Devices

**Jongin Choe [1], Taemin Lee [2] and Sanghyun Seo [3],***

[1] Department of Computer Science and Engineering, Chung-Ang University, Seoul 06974, Korea; jongin@cglab.cau.ac.kr

[2] Da Vinci Software Education Institute, Chung-Ang University, Seoul 06974, Korea; kevinlee@cglab.cau.ac.kr

[3] School of Computer Art, College of Art and Technology, Chung-Ang University, Anseong-si 17546, Korea

*** Correspondence: sanghyun@cau.ac.kr; Tel.: +82-10-7273-0318

**Abstract:** With the development of Internet technologies, chat environments have migrated from PCs to mobile devices. Conversations have moved from phone calls and text messages to mobile messaging services or "messengers," which has led to a significant surge in the use of mobile messengers such as Line and WhatsApp. However, because these messengers mainly use text as the communication medium, they have the inherent disadvantage of not effectively representing the user's nonverbal expressions. In this context, we propose a new emotional communication messenger that improves upon the limitations of existing static expressions in current messenger applications. We develop a chat messenger based on augmented reality (AR) technology using smartglasses, which are a type of a wearable device. To this end, we select a server model that is suitable for AR, and we apply an effective emotional expression method based on 16 different basic emotions classified as per Russell's model. In our app, these emotions can be expressed via emojis, animations, particle effects, and sound clips. Finally, we verify the efficacy of our messenger by conducting a user study to compare it with current 2D-based messenger services. Our messenger service can serve as a prototype for future AR-based messenger apps.

**Keywords:** augmented reality; emotional messenger; dynamic emotional expressions; smart devices; dedicated server model; 3D emojis

## 1. Introduction

With the worldwide prevalence of the Internet and the possibilities opened up by high-speed communication, multimedia chatting has emerged as a new and dominant form of communication. Today, the focus of IT has moved from PCs to smartphones and other mobile devices. With the increasing popularity of smartphones, smartphone-based messenger services have become indispensable for communication between users. Chatting in this mobile era centers around the mobile messenger application, which integrates the features of SMS/MMS into the instant messenger used in PCs. Figure 1 depicts an example of a typical mobile messenger application, whereas Table 1 lists the various messengers currently used.

Such messengers have begun expanding from PCs to mobile platforms and, consequently, the chat environment is constantly changing with the development of related technologies. However, while the mobile platform market has witnessed continual change, the messengers themselves have not changed significantly from the original text-based format in delivering information. The emotional expressions of existing messengers are static because of the 2D environment of the platform and, consequently, the messengers are limited in expressing 3D representations of user sensibility. Moreover, traditional

messengers cannot effectively represent the user's nonverbal emotional expressions because these services are based on text messaging. From Table 1, we note that none of the currently popular services offers the features of 3D-type messaging or nonverbal expressions.

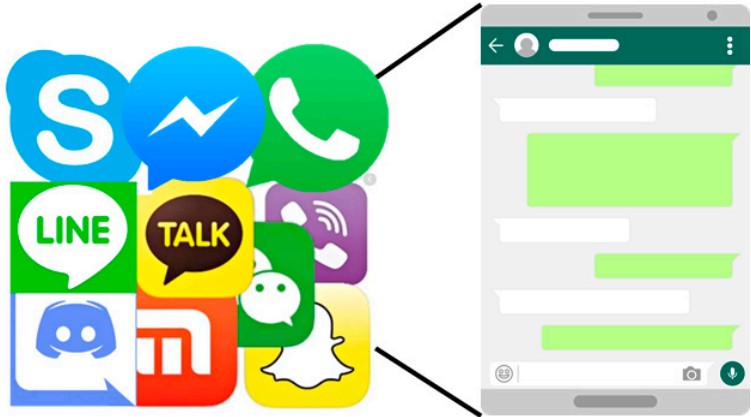

**Figure 1.** Typical mobile messenger application format (Skype, WhatsApp, Telegram, etc.).

**Table 1.** Popular messenger services and their available features [1].

|  | Monthly Active Users [2,3] | Emoji | 3D Message Chat | Themes |
|---|---|---|---|---|
| Discord | 250 M | Yes | No | Yes |
| Facebook Messenger | 1300 M | No | No | Yes |
| Skype | 300 M | No | No | No |
| Line | 203 M | No | No | Yes |
| Telegram | 300 M | No | No | Yes |
| WhatsApp | 2000 M | No | No | No |

In this context, we propose a new messenger that improves on the static presentation threshold of existing text-based messengers. The application offers a chat environment based on augmented reality (AR) technology with the use of wearable devices. Here, we note that wearable devices are used in virtual reality (VR) as well as AR technologies [4–9]. AR can improve on the shortcomings of the 2D environment and afford the use of dynamic and nonverbal expressions. In this study, users used smartglasses, a wearable device which facilitates the generation of an optimal user interface (UI) for an immersive experience [10,11]. In addition, we applied an effective emotional expression method based on various emojis to utilize the appropriate augmentation method for the smartglasses by utilizing the Vuforia AR software development kit (SDK). Subsequently, we selected a dedicated server model for the network environment required for the messenger and implemented the network model using application programming interfaces (APIs) provided by Unity. As the current 2D messenger is used on mobile phones, the inconvenience is unknown. If everyone carries a mobile phone, the use of 2D messengers can continue to be used. However, as the public use more smartwatches than ever before, it is certain that there will be an increase in the use of wearable devices. Wearable devices will be a 3D form of use interface, and we have conducted our research focusing on this part. Smart glass is expected to be the most commercial among 3D AR-enabled devices, so in this study, AR messenger can be used through smart glass.

This paper is organized as follows: Section 2 provides an overview of the studies related to our work. Section 3 presents the construction of our system; in particular, we describe our proposed emotional expression method and the messenger network design method. Section 4 presents the emotional expression results augmented by the proposed method. Finally, we discuss the performance of our messenger based on user studies. Section 5 presents the conclusions.

## 2. Related Work

Several previous studies have focused on existing messengers, their use, and specific aspects such as obtaining information or adding applications such as chatbots within the messengers [12–14]. However, current 2D messengers in their essential form limit users from expressing complex emotional feelings, and very few studies have focused on these aspects. Consequently, we propose 3D-based messaging with the application of AR.

AR as a technology can be considered to have arrived with the head-mounted display (HMD) in 1968. This device was the predecessor to current see-through smartglasses that use a translucent display [15], and the invention of HMDs has subsequently led to intensive AR studies [16,17]. In particular, many studies have focused on the provision of information through AR. In this regard, Shu et al. proposed an AR-based social network framework in which users can immediately disseminate and view information from others. The study demonstrates the possibility of how users can communicate/advertise messages as well as transmit facial expressions to all nearby devices via device-to-device communication. When a user sights another person through a wearable device that can be used as a camera, the framework automatically extracts the facial features of this person of interest, compares them to previously captured features, and provides the user the information shared by the other user [18]. In this regard, Bâce et al. extended the concept of AR tags or annotations to a ubiquitous message system based on an eye gesture that locks the message to the distinguishing characteristics of an actual object. This study proposed a social interaction approach in which a user can store AR messages onto real objects, and these messages reappear when a second user uses a specific gesture to retrieve information [19].

While research on 3D messengers has not been done much, there have been several studies of 3D avatars and 3D emotional expressions. A study by Ann J. et al has investigated the visualizing changes in users' emotional states. The user's emotional information was inputted and printed out with the avatar's facial expression [20]. A similar study can be seen by Ronan B. et al, which also suggests how to efficiently map facial expressions to them based on the arousal-valence emotional model [21]. Both studies have conducted 3D representation of emotions, but because our research is an avatar expression on smart devices, it is different from these studies that are expressing avatar emotions on PCs. Also, the avatar's emotional expression is one of the advantages of 3D messenger, and our research has a big difference: the use of messenger on a wearable glass. Most of the abovementioned studies have used AR in conveying information only undirectionally. Our research has two different features compared with other applications. First, emotional expression that can use instant messaging. This is the biggest difference from traditional 2D messengers. For this purpose, a messenger using AR was used. Second, it provided various functions by focusing on a messenger through smart devices. Existing AR technologies focused on providing visually pleasing information, such as games, but we made it possible for users to communicate well with each other.

## 3. Materials Methods

### 3.1. System Concept

We developed an AR-based messenger that is more diverse and intuitive than current 2D messengers, which can only transmit simple sentences and emoticons. Our AR messenger, in conjunction with smartglasses, allows users to add emotions to their thoughts and feelings through a new level of expression vocabulary, which can increase the complexity level of communication among users. Our AR messenger also allows multiple users to communicate in chat rooms through their smartglasses, smartphones, and PCs. In this study, we repurposed existing network connection methods to enable our messenger to facilitate communication between users, and we also developed suitable emotional expressions for our AR messenger. Figure 2 shows the conceptual diagram of our emotional messenger. As we can see in Figure 2, many users can access to virtual AR space using

their devices such as smartglasses, tablet, and personal computer through a dedicated server. In AR messengers, they can describe their emotions by various expressions of avatars.

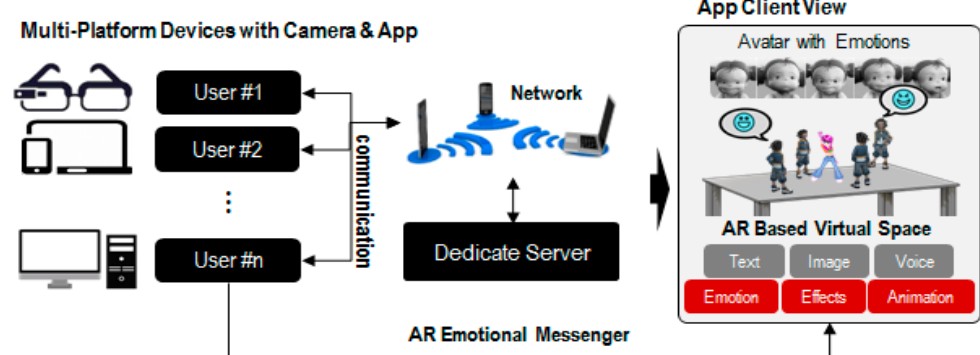

**Figure 2.** Concept of augmented reality (AR) emotional messenger based on smartglasses and other devices through a dedicated server.

### 3.2. Emotional Expressions

Traditional messengers only allow communication via text and 2D images. In such applications, there is a limit to the delivery of user emotions because only static information is available for communication. This limitation has been overcome to a certain extent in previous 2D messengers via delivering 2D-animation-based emotions. However, such expressions are limited relative to 3D emotional expressions such as 3D emojis. In our case, we enable the use of emotional expressions in the AR environment, which are analyzed and applied to 3D objects through the messenger. Our approach uses eight expressions, which were selected based on Russell's emotional model [22], and eight other expressions commonly used in regular messengers. The selected expressions are listed in Table 2. In our messenger service, 16 different emotional expressions can be expressed in four different ways to convey different emotions.

**Table 2.** Emotions and expressions used in our approach.

| | |
|---|---|
| Emotions based on Russell's model | Anger, Excitement, Happiness, Satisfaction, Tiredness, Gloom, Frustration, Pain |
| Frequently used visual expressions | Hello, Thanks, Sorry, Congratulations, Embarrassed, No, Aggressive, Pointing |

In our approach, various emojis allow users to present 3D, dynamic representations of emotions and effectively communicate the information to others. Furthermore, four emotional expression methods were developed to allow enhanced character expression in 3D (Figure 3). Here, we note that humans express or recognize emotions nonverbally through behavior, facial expressions, and vocal intonation [23]. Furthermore, feelings and intentions during the course of communication are also understood via action or body language. In this regard, some studies have reported that humans use a series of stereotypical gestures expressed according to their emotions [24,25]. For example, the expression of joy generally takes on the form of gestures such as the drooping of the corners of the eyes, lifting of the corners of the mouth, forceful fisting of both hands, and opening of the chest and stretching of both arms. On the contrary, grief generally corresponds to the gestures of covering the mouth or face to prevent crying. Consequently, in our approach, the actions corresponding to the human expression of emotions were synthesized with 3D animation so that the avatars in the AR environment could perform them in the messenger, along with using emojis corresponding to each emotion.

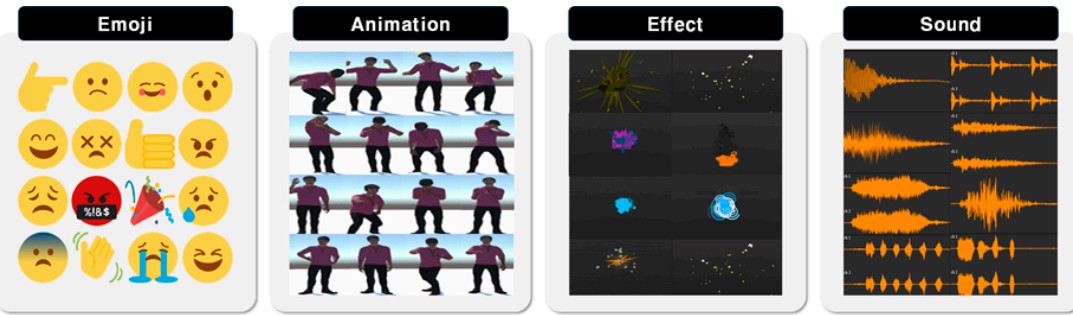

**Figure 3.** Four types of expression available in our messenger service.

### 3.3. Development of Messenger Network

We used the dedicated server networking model provided by UNet to create a messenger chat environment. UNet is a service that aids in building network infrastructure with high-level APIs supported by Unity [26,27]. In UNet, high-level script APIs allow for the organization of networks without the need for considering low-level details to easily meet common requirements.

In our study, we selected dedicated server, a system that accommodates all clients connected to the server. Furthermore, the system also enables synchronization between clients. We chose Dedicated Server as our model for the following reasons: First, logs such as user connection status and communication status can be identified and managed at one location. Second, the model can detect and address incorrect information or unexpected problems. Third, it can accommodate a large number of messenger users. We used this server to process the types of information listed in Table 3.

**Table 3.** Information processing/synchronizing capabilities of Dedicated Server.

| Information Processing/Synchronizing Capabilities of Dedicated Server |
| :---: |
| Assignment of client ID |
| Message transmission and synchronization |
| Management of client's entry and exit |
| Creation of chat rooms and updating of lists |

The process of information synchronization is shown in Figure 4. Here, clients can refer to any type of user interface to the messenger: smartphone, smartglasses, PC, etc. The server uses a PC as the host, and one or more servers are required for user access to the messenger. The server must be accessed by a limited number of people to handle the synchronization processing for each client. Messages are transmitted through a serialization process to ensure that the platform is correctly rendered in other computing environments, and a deserialization process is performed in the course of receiving messages. Chat messages, which are sent between clients and servers, are basically in the form of text, images, and voice notes, and character behavior, particle effects, and sound effects are also available for emotional expression.

Because the amount of chat messages increases proportionally with the number of participants in chat rooms, it is necessary to set data-size restrictions to avoid communication delays. UNet recommends a total of 1500-byte data packets for transmission between mobile devices; usage beyond this limit can lead to packet loss or unstable transmission due to network communication failure. Therefore, in our study, we limited the packet data to 1400 bytes. The quality of service (QoS) used for client-to-server transmission could then be set as reliable and sequenced. The size of images to be transmitted was reduced via PNG compression, and if the packet size exceeded the limit, the image was segmented and sent over a number of small packets. This method of transferring images in segments is presented in pseudocode in Algorithm 1.

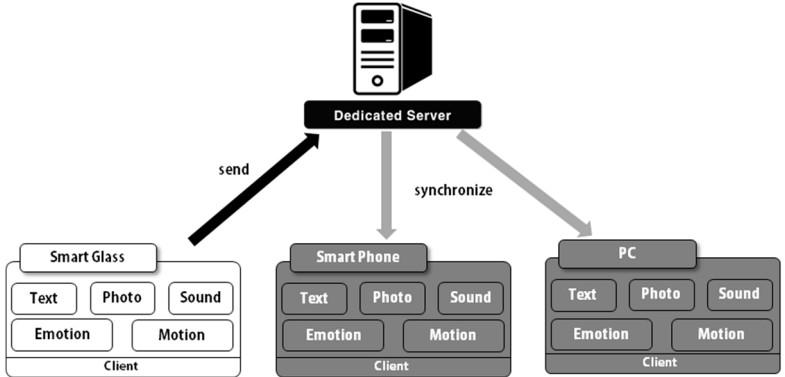

**Figure 4.** Process of synchronization by dedicated server (black arrow: transmission messages to server from client, gray arrows: print messages received from server on client screen).

---

**Algorithm 1** Segmenting and transmitting images (*Italics* for comments)

---

1:    **function** SendTextureData ( Texture T ) {
2:    copyData = EncodeToPNG(T)                *// Copy compressed T into PNG*
3:    sndBuffer: = new byte[1400]             *// Set the size of transmission buffer to 1400 bytes*
4:    **for** i, pos = 0 to copyData.Length {
5:       sndBuffer[i] = copyData[pos]          *// Save image data to transmission buffer*
6:       **if**( i > sndBuffer.Length && pos > copyData.Length ) {   *// If the transmission buffer is full and the partitioned packet is the last one, send and exit system*
7:           SendPacket(sndBuffer)
8:           **break**
9:       }**else if**(i > sndBuffer.Length){   *// Transfer if the transfer buffer is full, but not the last split packet*
10:         SendPacket(sndBuffer)
11:         i = 0
12:      }**else if**(pos > copyData.Length){

          *// If the transfer buffer is not full but the split packet is the last one, then exit after transfer*

13:         SendPacket(sndBuffer)
14:         **break**

      }
   }

---

To build a chat environment using AR, we used the marker detection method. Using markers to create an environment is very familiar in the area of AR [28,29]. When users trigger markers through their smartglasses, they can automatically participate in the chat messenger. Therefore, markers act as chat rooms in the messenger and have a structure in which clients participate in chat rooms created on the server. Each marker corresponds to a separate chat room, and the server also performs synchronization processing with the data management of different users.

## 4. Results

In our study, we used the EPSON smartglass MOVERIO BT-300. We implemented AR using the Vuforia SDK and Unity game engine to ensure that our messenger service was compatible with smartglasses-supporting Android operating systems, and we built into android application package file (APK) for demonstration. When the natural user interface (NUI) is supported through smartglasses, users recognize markers on the camera built into the smartglass without learning an operating key. They

read the information of the marker to augment the maps and characters in the chat room corresponding to the marker. The application is designed to be used not only in AR through smartglasses, but also in mobile devices, smartphones, and PCs. Figure 5 shows the messenger environment on a variety of devices.

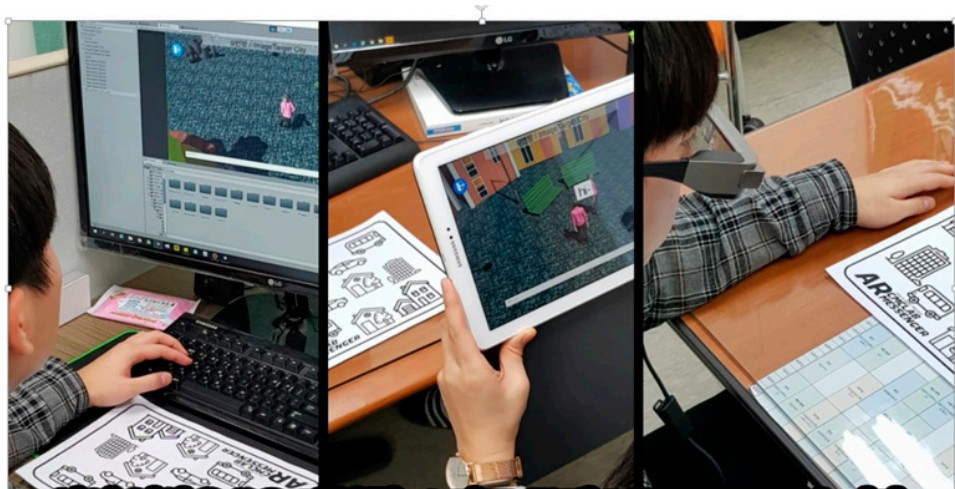

**Figure 5.** Messenger execution on a variety of devices.

Figure 6 shows an augmented screen based on the marker information when the user identifies a marker through the smartglass. The UI displays the name of the chat room to which the user is currently connected, at the right portion of the screen, and emojis are available to the user at the screen's center-right. Other functions such as text, image, and voice recognition buttons are provided at the bottom of the screen and the top-left of the chat room buttons, following the layout of currently popular messengers to ensure user familiarity.

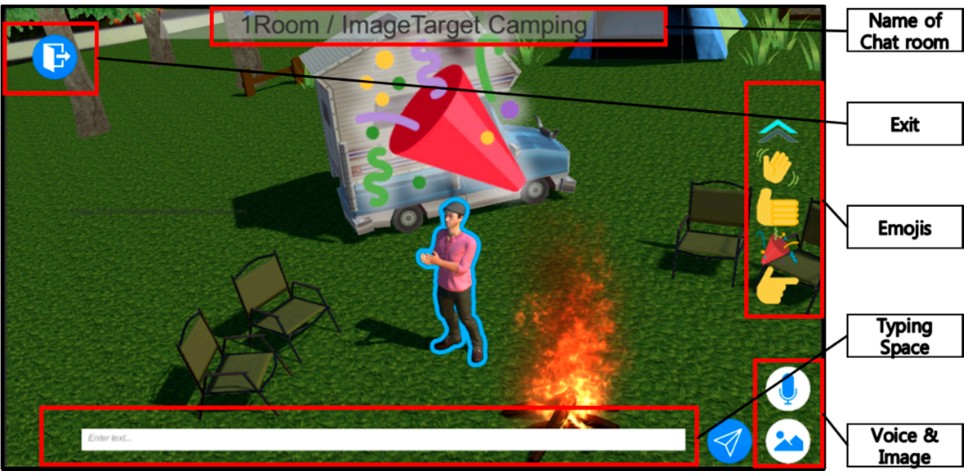

**Figure 6.** Augmented chat room and characters viewed through marker.

Our application provides chat rooms with different types of map for each marker. There are two types of chat room markers used in the messenger, and each time a marker is created, the map of the chat room can be expanded. The design of each marker is shown in Figure 7. Figure 7a shows images of buildings and cars representing cities, and Figure 7b shows images of tents and trees representing camping sites. These markers are designed to provide users an intuitive understanding of the concept of the chat rooms available in our application.

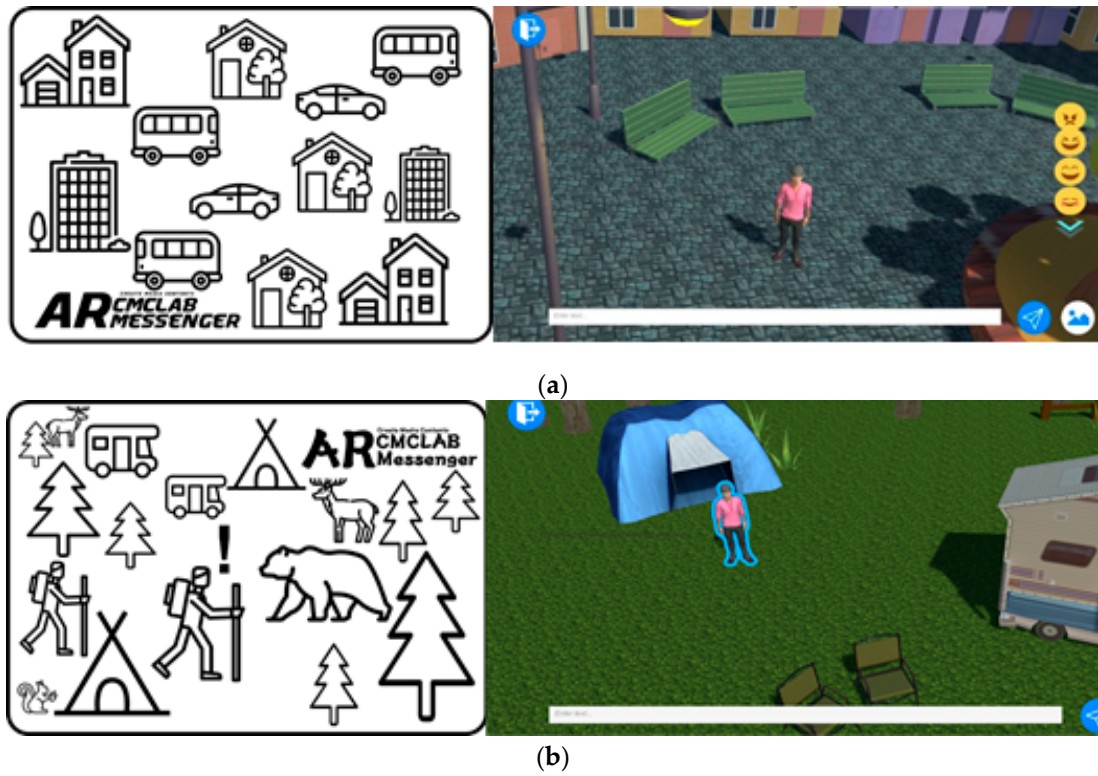

(**a**)

(**b**)

**Figure 7.** Maps with markers provided by our application. (**a**) City map and marker. (**b**) Camping map and marker.

The attributes of 3D space can lead to certain problems in the functioning of the AR messenger; texts become hidden or visible depending on the camera direction in which the object is viewed by the character. To solve this problem, we ensured that 2D objects such as speech bubbles and text continue facing the camera so that text, images, etc. can be visible to users along any direction without blocking or destroying other objects. Figure 8 shows how 2D objects (text) are printed and displayed to the user in a 3D environment.

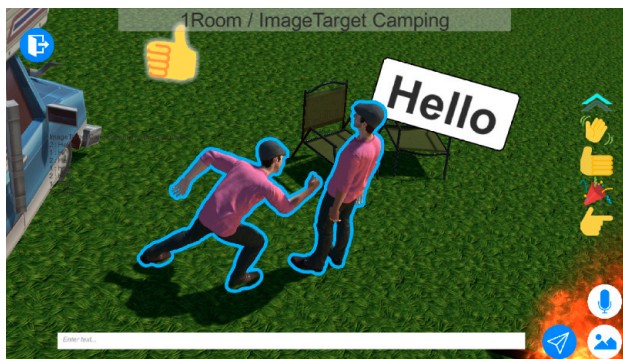

**Figure 8.** Adjustment of display result considering camera position.

When a user enters a chat room using the marker shown in Figure 7, the maps and characters are augmented with other players connected to this chat room. The user can communicate with other users through emojis available on the layout panel on the right and the chat window and buttons at the bottom. Image buttons can be linked to galleries on the user device to retrieve and transmit images within them. The application also allows for the use of Google Voice Recognition to provide a mobile environment wherein voice recognition can be employed.

Users can share and communicate emotions with other players through a character by simply tapping on the emojis placed on the center-right side of the screen. The user avatar can express emotions through emojis in various ways with action, sound, and other effects. Figure 9 shows a user communicating via an avatar that expresses the emotion corresponding to the emoji chosen by the user.

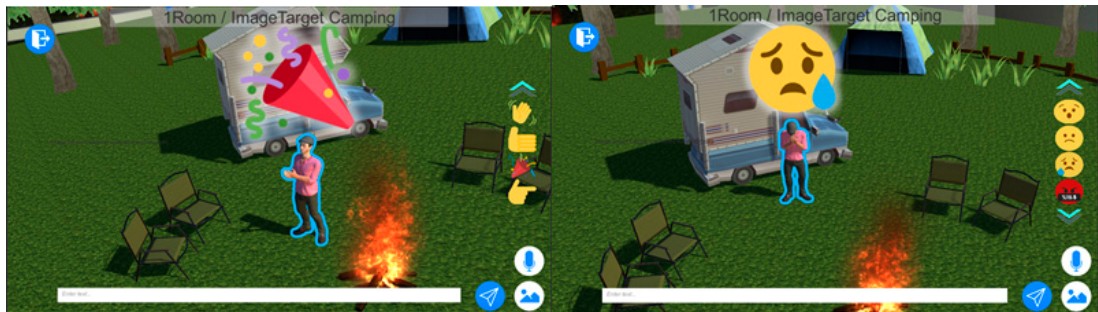

**Figure 9.** Character expressing emotions through emojis.

In this study, various ways of expressing opinions existing in existing messengers were expressed in synchronizing with augmented reality space. Smart glass has different input/output methods from existing devices, so it allows users to type characters through voice recognition. For keystrokes of traditional AR glass (see Figure 10a), there may be disadvantages of being uncomfortable or slow when users type. We did most of the methods with buttons, and in the case of chats requiring keystrokes, we added the ability to convert voice data into text through voice recognition. At this time, Google Cloud Speech" was used, which had a high recognition rate among existing APIs. Like Figure 10b, if we recognize the voice in Google Voice Recognition window, we can conveniently message the words we want to AR environment without typing.

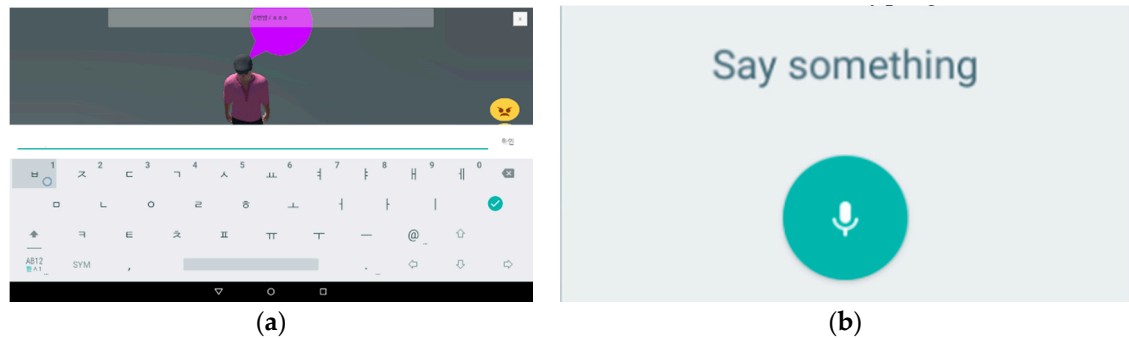

(**a**)  (**b**)

**Figure 10.** The methods for sending messages in AR messengers; (**a**) is using traditional keystrokes, (**b**) is voice recognition using Google Cloud Speech [30].

Figure 11 shows various instances of how users in the same chat room can communicate through our messenger. We note that emojis are clearly visible regardless of their position and that a variety of emotional expressions (reflecting the user's feelings) can be visually portrayed. The application also allows three or more simultaneous interactions as opposed to only sequential communication available in 2D text messengers.

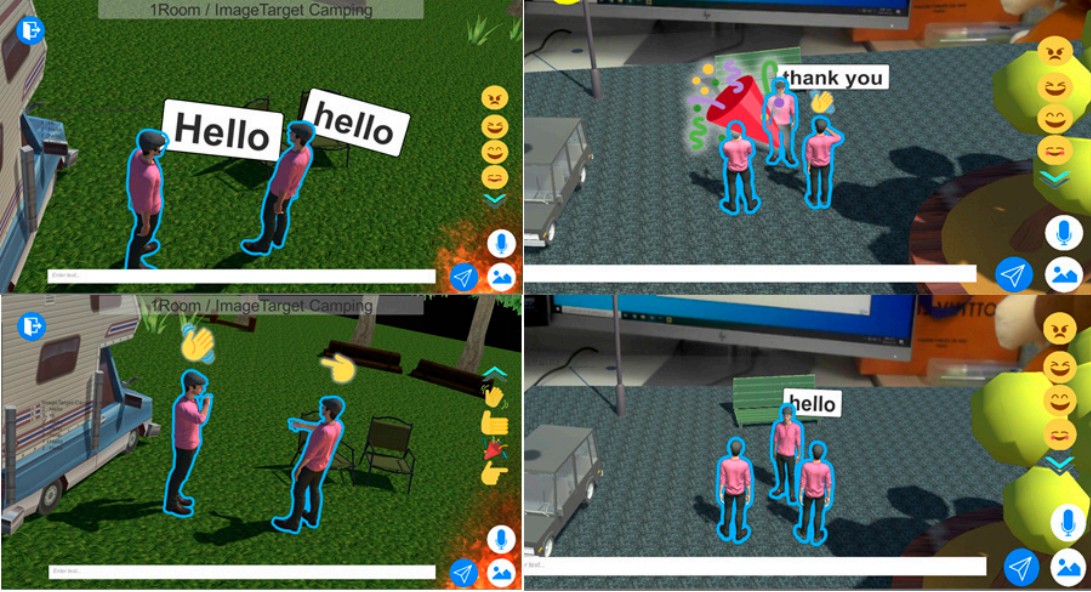

**Figure 11.** Dynamic communication through messenger.

## 5. Discussion

We conducted a user study to compare the performance of our messenger service relative to that of 2D messengers. The users comprised 35 students who were in their 20s and were asked four questions each. These students were chosen because individuals in this age group form the dominant section of messenger users relative to other age groups. The four questions were: 1) Is the messenger convenient to communicate with other people? 2) Is the messenger dynamic when compared with the other messengers based on 2D text? 3) Would you replace the 2D messenger you are using now? 4) Does it seem appropriate for the next generation of messengers? Users could assign scores ranging from 1 to 5, with 1 being the lowest score and 5 the highest. Figure 12 shows the results of our user study. From Figure 12a, we note that a majority of users assigned a score of 4, while the average score was 3.57. This value is above average, which indicates that our messenger is user-friendly. From Figure 12b, we note that most users found our messenger to be better than 2D messengers. The average score of 4.05 indicates that users feel that the application is dynamic in communicating with other users. If (a) and (b) were surveys of the usability of simple 3D messengers, Figure 12c was a survey of whether existing 2D could be replaced. However, Figure 12c's evaluation did not seem to replace 2D messengers at the moment, as it received a lower rating. However, through Figure 12d, our messenger has shown us the potential for future development as the next messenger. Rather than competitiveness in the present, where smartphones are at the center, our messenger in the future, especially on wearable devices, can certainly be seen as an advantage.

Our research is the first to develop a 3D AR messenger using smart glass. We would like to compare our contributions with some similar studies. The biggest difference from the existing 2D messenger is that 3D chatting through avatars is possible. While today's messenger may be more efficient to chat over mobile phones, the time will come for 3D chatting to become more convenient as more wearable devices evolve [31,32]. Also, as 3D chatting is done, the theme of the chatting background, emoji that can express emotions in three dimensions, can be used. Similarly, in the case of [33], who studied emotional avatars, the expression was expressed using AV value in emotional expression. There exists a similar part in avatar's emotional expression to our research, but this is focused on emotional expression. Our research is different in that it is a messenger that enables various communication between users. Compared to Avatar Chat [34], our research is done through smart glass. If a user enters a chat room directly through a smart glass, he or she can enter the avatar and chat more intimately, which can arouse interest. The research on emoji [20] is ultimately more likely

to be included in our messengers and developed. [20] is different from our research because it is not conducted on smart devices, although it expresses emotion by reading the user's face. As a result, we developed 3D emotional messengers through wearable devices, especially smart glass that users can immerse in, which is a future-oriented messenger.

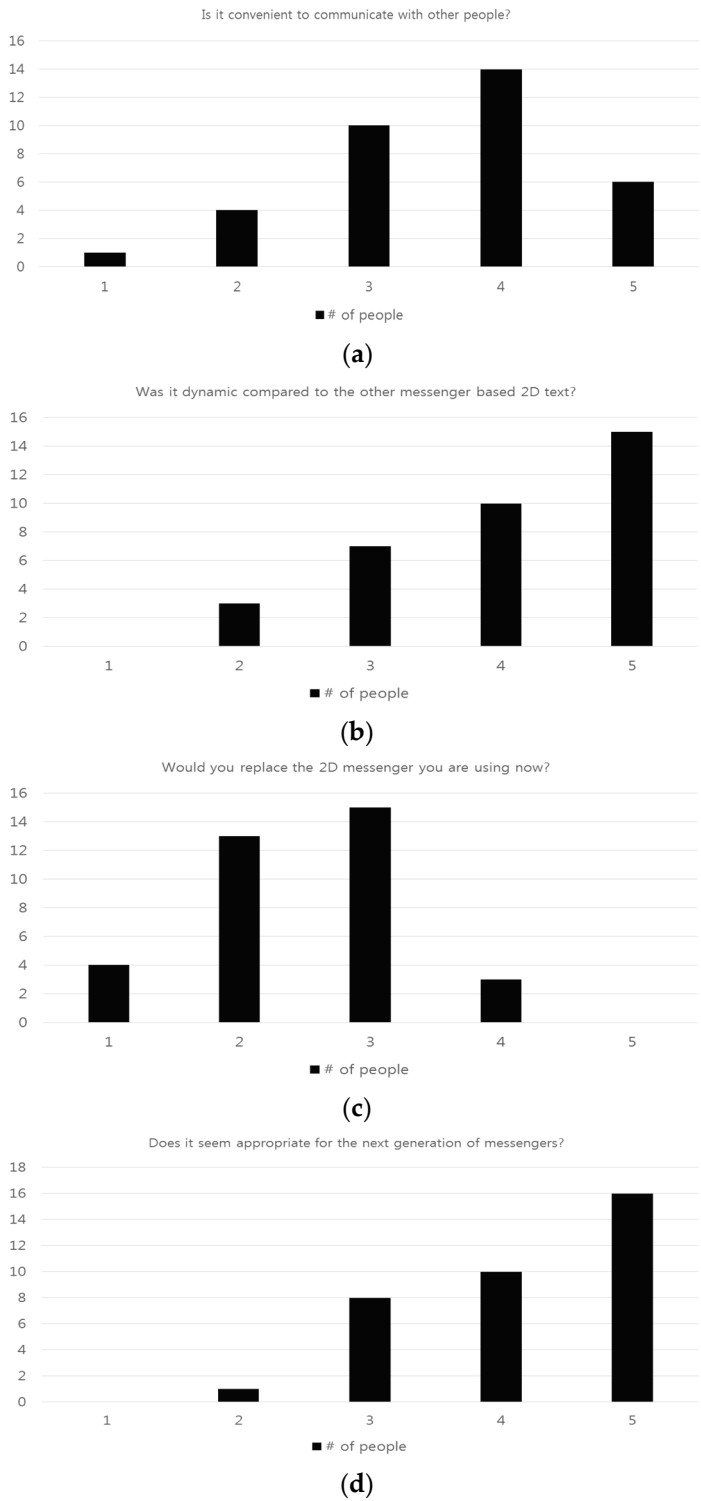

**Figure 12.** User validation of proposed messenger based on the following questions: (**a**) Is the application convenient to communicate with other people? (**b**) Is the application dynamic relative to other messengers based on 2D text? (**c**) Would you replace the 2D messenger you are using now? (**d**) Does it seem appropriate for the next generation of messengers?

These big differences can be found in Figure 13. Our system has overwritten the advantages of AR emotional avatars based on messengers. As we can see in Figures 13a and 13b, a large number of people, not two or three people, come into the room and express their feelings or chats at the same time. It can be seen that this is different from other AR emotional avatar technologies. The general emotional avatar technology [34] can only be seen as expressing emoji to the opponent, but our research shows a large number of people interacting with each other and forming messengers. Also, in Figure 13c, users can express their identity through a variety of characters, not the same characters. Currently, it is the line where users can choose a few characters before entering the room, but this part gives the user a choice and he or she enters AR messenger through the character, allowing him or her to become more immersed in the chat room.

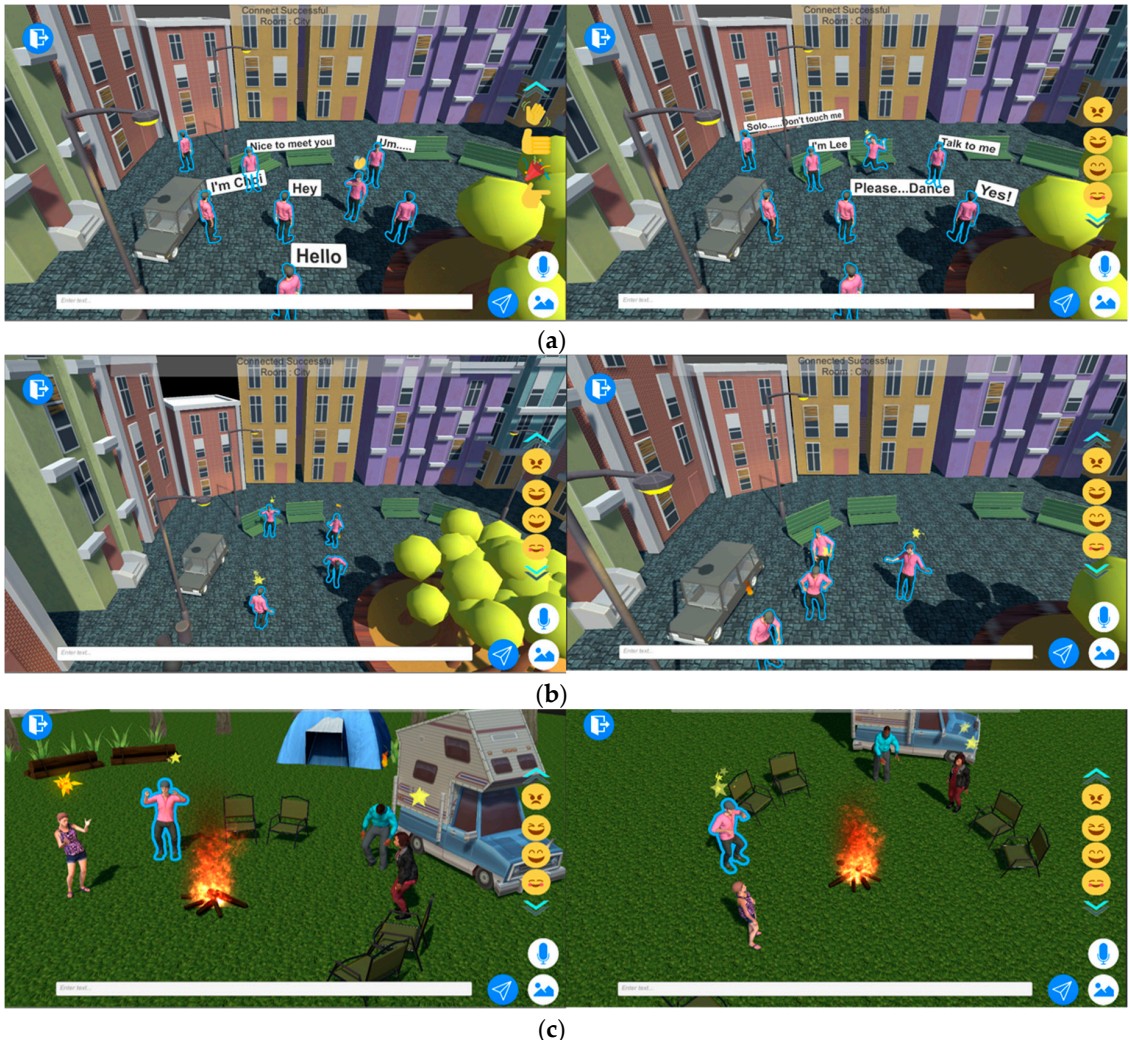

**Figure 13.** Advantages of AR messenger in our system: (**a**) eight users are accessing at the same time and chatting at the same time. (**b**) Four users are expressing their feelings using emoji. (**c**) Each person is participating in chatting using a different avatar.

## 6. Conclusions

We proposed an AR-based emotional messenger to overcome the static conversation constraints of existing text-based messengers. The highlights of our study are as follows: First, our utilization of smartglasses allows users to experience AR and effectively immerse themselves in the chat environment. The use of messenger using smart glass is expected to increase in usability as IT devices develop in the future. It also provides a wider field of view compared to smartphones, so it can prevent problems

that may arise from smartphones. For example, an accident occurs while walking without looking forward while using a messenger with a smartphone. It can also be said that by creating messengers based on smart glass, it provides a platform for adding various contents. The various games and applications that currently exist can be expanded to run on top of our AR messengers. Second, we used the Google Voice Recognition API to provide improved conversational flexibility over conventional and cumbersome tapping-type input methods. Third, we designed several tracking techniques to print virtual objects on markers that were designed using marker detection technology. These markers are designed to infer their chat room maps so that users can access them intuitively. The advantage of creating maps using markers is that they can be easily expanded. Fourth, emotions could be expressed through emojis. To this end, we used a 2D emotional model established by Russell, and each of the 16 universal emotions selected in the emotional model were augmented by animation, particle, and sound effects to enhance emotional expression. This aided users in expressing their feelings in a 3D and dynamic manner. Fifth, we applied the dedicated server model to facilitate the transmission of chat messages between servers and clients, and the server provided detailed processing of chat messages that were synchronized. Finally, we examined the user experience of the messenger. These results indicate that users can differentiate and enjoy the benefits of 3D over 2D. The AR emotional messenger proposed in this study can serve as a basic model for future AR applications as well as AR content research. We also believe that our messenger can be a representative example of the next generation of instant messaging services based on the use of smartglasses.

In this study, virtual characters were used as tools to express emotions in AR. In future, we plan to enable users to customize their faces. Customization possibilities can involve insertion of the user's face on the character's face, as in the case of Mii (Nintendo Cooperation), or using expressions that synthesize a user's picture in their character in other games. This can aid in more direct AR experience by users. In addition, we used marker detection technology to implement AR, which affords scalability with increase in the map size; however, the technology does not cover all objects in daily life. In other words, it is currently difficult to generate results that are close to the complexity of the real world. Therefore, we believe that when AR maps based on the real world are available for all users of the same chat room, more effective content can be produced.

**Author Contributions:** J.C., T.L. and S.S. participated in all phases and contributed equally to this work. T.L. wrote the paper and contribute on results and discussion. T.L. and S.S. revised the whole paper. J.C. developed messenger and T.L. performed experimental data collection and S.S. advised in the process of paper-writing. All authors have read and agreed to the published version of the manuscript.

**Funding:** This work has supported by the National Research Foundation of Korea (NRF) grant funded by the Korea government(MSIT) (No. NRF-2019R1F1A1058715) and supported by the Chung-Ang University Research Scholarship Grants in 2019

**Acknowledgments:** This work has supported by the National Research Foundation of Korea (NRF) grant funded by the Korea government(MSIT) (No. NRF-2019R1F1A1058715) and supported by the Chung-Ang University Research Scholarship Grants in 2019

**Conflicts of Interest:** The authors declare no conflict of interest.

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
