# Peer review of "Augmented-Reality-Based 3D Emotional Messenger for Dynamic User Communication with Smart Devices"

_electronics, doi:10.3390/electronics9071127_

Round 1
Reviewer 1 Report
This is a very interesting idea about the implementation of a virtual messenger system where users can send their emotions through emojis that are displayed near their avatars. The evoloution of graphics technology along with the increasing use of messenger systems favors the implementation of similar systems.
The paper is well written and is accompanied by adequate references. The exploitation of well established emotion libraries is also welcome.
The authors state that this system is an AR-based. The use of the system is evaluated positively by end users.
However, the exploitation of AR is very limited throughout the use of the application with AR glasses (an not with a tablet, or a PC) for just activating the messenger room. While reading the introduction, one could anticipate a more thorough integration of AR throughout the operation of the system, especially wjen dealing with the expression of emotions. For instance, Samsung AR emojis capture the expression of a user and turns into animation through AR. This is a far more sophisticated use of AR for capturing emotions than just tapping on a set of icons. The overall idea of using AR glasses (which is not the most convenient interaction medium!) for just initiating the chat room does nor seem to be appealing. If AR is exploited in some other way, at least it is not clear with the provided description.
The use of avatars and expressions over them through tablet as also descibrd in the paper is a more straightforward approach and maybe the authors should just focus there than trying to enrich the interface with AR features that are scarcely exploited.
Therefore, the authors are proposed to reorganize the notion of the contribution of their paper and focus on the development of the interface inside the messenger rooms - there lay some intereseting ideas. They should compare their idea with older generic interfaces such as Second World etc. They also should expand the number evaluaters. Finally, they should divide clearly the design and implementation of the application from the use of interface and the evaluation.
Author Response
1) The exploitation of AR is very limited throughout the use of the application with AR glasses (an not with a tablet, or a PC) for just activating the messenger room(Contribution).
Answer: We explained the contribution of our research and thought that the advantages of it were shown through validation. However, the validation seemed to be lacking, so the user-study was further reinforced and added. That part is added to the latter part of Chapter 5, and a description of the contribution is added to the lower part. That part is as follows.
Part of chap 1. In this context, we propose a new messenger that improves on the static presentation threshold of existing text-based messengers. The application offers a chat environment based on augmented reality (AR) technology with the use of wearable devices. Here, we note that wearable devices are used in virtual reality (VR) as well as AR technologies [1, 2, 3, 4]. AR can improve on the shortcomings of the 2D environment and afford the use of dynamic and nonverbal expressions. In this study, users used smartglasses, a wearable device, which facilitates the generation of an optimal user interface (UI) for an immersive experience[5, 6]. In addition, we applied an effective emotional expression method based on various emojis to utilize the appropriate augmentation method for the smartglasses by utilizing the Vuforia AR software development kit (SDK). Subsequently, we selected a dedicated server model for the network environment required for the messenger and implemented the network model using application programming interfaces (APIs) provided by Unity. As the current 2D messenger is used on mobile phones, the inconvenience is unknown. If everyone carries a mobile phone, the use of 2D messengers can continue to be used. But just as smartwatches use more smartwatches than ever before, it is certain that there will be an increase in the use of wearable devices. Wearable devices will be a 3D form of use interface, and we have conducted our research focusing on this part.
Part of chap 6. We proposed an AR-based emotional messenger to overcome the static conversation constraints of existing text-based messengers. The highlights of our study are as follows: First, our utilization of smartglasses allows users to experience AR and effectively immerse themselves in the chat environment. The use of messenger using smart glass is expected to increase in usability as IT devices develop in the future. It also provides a wider field of view compared to smartphones, so it can prevent problems that may arise from smartphones. For example, an accident occurs while walking without looking forward while using a messenger with a smartphone
|
2) They should compare their idea with older generic interfaces such as Second World etc. They also should expand the number evaluators. Finally, they should divide clearly the design and implementation of the application from the use of interface and the evaluation (User study & Validation).
Answer: We have verified our results by adding the number of people in the existing user study and adding two other surveys. The added verification methods and contents are added to Figure 11 and are as follows.
We conducted a user study to compare the performance of our messenger service relative to that of 2D messengers. The users comprised 35 students who were in their 20s and were asked 4 questions each. These students were chosen because individuals in this age group form the dominant section of messenger users relative to other age groups. The four questions were: 1. Is the messenger convenient to communicate with other people? 2. Is the messenger dynamic when compared with the other messengers based on 2D text? 3. Would you replace the 2D messenger you are using now? 4. Does it seem appropriate for the next generation of messengers? Users could assign scores ranging from 1 to 5, with 1 being the lowest score and 5 the highest. Figure 11 shows the results of our user study. From Figure 11(a), we note that a majority of users assigned a score of 4, while the average score was 3.57. This value is above average, which indicates that our messenger is user-friendly. From Figure 11(b), we note that most users found our messenger to be better than 2D messengers. The average score of 4.05 indicates that users feel that the application is dynamic in communicating with other users. If (a) and (b) were surveys of the usability of simple 3D messengers, Figure 11(c) was a survey of whether existing 2D could be replaced. However, (c)'s evaluation did not seem to replace 2D messengers at the moment, as it received a lower rating than However, through Figure 11(d), our messenger has shown us the potential for future development as the next messenger. Rather than competitiveness in the present, where smartphones are at the center, our messenger in the future, especially on wearable devices, can certainly be seen as an advantage.
|
Reviewer 2 Report
This paper presents a system of virtual or augmented reality for social media communication. The system seems to involve users looking into a virtual chat environment and then using animated characters to represent emotional states in that environment. This is claimed to be superior to standard 2D messaging environments such as Facebook Messenger, etc. The paper seems to be primarily about the architecture of the system with some details on the types of emotional content displayed. I find it difficult to see this work as an original and novel contribution to the field. Expressive 3D avatars are well-known in 3D gaming and providing users with the ability to control such avatars in the context of a chat environment, doesn't appear to be sufficiently novel or original to merit publication. The authors compare their system to 2D chat systems in an experiment that is not terribly comprehensive. The subjects are asked whether they prefer the proposed system over 2D systems, but no particular 2D system is chosen for comparison and the subjects do not have structured tasks to perform. The authors ignore the benefits of 2D systems such as the ability to chat while on public transport or respond to messages quickly on small devices like smart phones. They also ignore other 3D environments that are routinely used for chat and social interaction such as some gaming environments. For example Minecraft, Counterstrike or any other number of 3D gaming systems that allow users to come together in a 3D environment for social interaction and messaging. It is unclear about whether the presence of emotional representation of avatars in this environment is helpful. The authors should widen the scope of their experimental comparisons to include 3D environments that are used for messaging, not just comparing 3D to 2D only. In addition, there is other research in the field of 3D chatting environments with emotional representation that the authors should consider and place their work in context against such as: Ahn, Junghyun & Gobron, Stéphane & Garcia, David & Silvestre, Quentin & Thalmann, Daniel & Boulic, Ronan. (2012). An NVC Emotional Model for Conversational Virtual Humans in a 3D Chatting Environment. 7378. 47-57. 10.1007/978-3-642-31567-1_5. Since chatting in 3D environments is a known technology, the authors need to expand their experimental justification to cover and compare the proposed method with 3D environments and the above work (and others may also exist). Comparison with 2D environments with simple emoji representation does not show the benefits of the proposed 3D emotion representation system sufficiently in my opinion. The 2D environments still have considerable advantages in terms of simplicity of use that are not considered in the very unstructured experimental justification that the authors provide. If the authors wish to focus on the advantages of their particular emotional representations, they need to describe them more clearly in the paper and perform a more rigorous experiment, perhaps asking participants to perform set tasks with and without the 3D emotional representations and then asking the users about how accurately they felt the emotions of the other people in the chat room or how well they sympathized with the others in the chat room. The experimental justification of this paper is far too simple to support the conclusions. There is no evidence presented that the emotional representations are key to the user's enjoyment of the environment. The 3D environment itself may be more fun to play in, but since there are other 3D chat environments available, then there doesn't seem to be much benefit to the proposed approach. The authors need to greatly improve the experimental justification before future publication.Author Response
1) Better than 2D mobile phone messenger?
Answer: As you said, I don't think we can say that our messenger is more usable than our current messenger. Clearly, 2D messengers using mobile phones have advantages of ease of use or portability. But that's the current situation. Just as many people use smartwatches, more and more devices are being used on smartphones. As a result, the use of wearable devices is expected to surpass mobile phones. It is a next-generation messenger in preparation for that time. I think our research has suggested the potential for future development rather than the current situation.
2) I find it difficult to see this work as an original and novel contribution to the field (Contribution).
Answer: We explained the contribution of our research and thought that the advantages of it were shown through validation. However, the validation seemed to be lacking, so the user-study was further reinforced and added. That part is added to the latter part of Chapter 5, and a description of the contribution is added to the lower part. That part is as follows.
Part of chap 1. In this context, we propose a new messenger that improves on the static presentation threshold of existing text-based messengers. The application offers a chat environment based on augmented reality (AR) technology with the use of wearable devices. Here, we note that wearable devices are used in virtual reality (VR) as well as AR technologies [1, 2, 3, 4]. AR can improve on the shortcomings of the 2D environment and afford the use of dynamic and nonverbal expressions. In this study, users used smartglasses, a wearable device, which facilitates the generation of an optimal user interface (UI) for an immersive experience[5, 6]. In addition, we applied an effective emotional expression method based on various emojis to utilize the appropriate augmentation method for the smartglasses by utilizing the Vuforia AR software development kit (SDK). Subsequently, we selected a dedicated server model for the network environment required for the messenger and implemented the network model using application programming interfaces (APIs) provided by Unity. As the current 2D messenger is used on mobile phones, the inconvenience is unknown. If everyone carries a mobile phone, the use of 2D messengers can continue to be used. But just as smartwatches use more smartwatches than ever before, it is certain that there will be an increase in the use of wearable devices. Wearable devices will be a 3D form of use interface, and we have conducted our research focusing on this part.
Part of chap 6. We proposed an AR-based emotional messenger to overcome the static conversation constraints of existing text-based messengers. The highlights of our study are as follows: First, our utilization of smartglasses allows users to experience AR and effectively immerse themselves in the chat environment. The use of messenger using smart glass is expected to increase in usability as IT devices develop in the future. It also provides a wider field of view compared to smartphones, so it can prevent problems that may arise from smartphones. For example, an accident occurs while walking without looking forward while using a messenger with a smartphone
|
3) The authors need to greatly improve the experimental justification before future publication (Validation).
Answer: We explained the contribution of our research and thought that the advantages of it were shown through validation. However, the validation seemed to be lacking, so the user-study was further reinforced and added. That part is added to the latter part of Chapter 5, and a description of the contribution is added to the lower part. That part is as follows.
We conducted a user study to compare the performance of our messenger service relative to that of 2D messengers. The users comprised 35 students who were in their 20s and were asked 4 questions each. These students were chosen because individuals in this age group form the dominant section of messenger users relative to other age groups. The four questions were: 1. Is the messenger convenient to communicate with other people? 2. Is the messenger dynamic when compared with the other messengers based on 2D text? 3. Would you replace the 2D messenger you are using now? 4. Does it seem appropriate for the next generation of messengers? Users could assign scores ranging from 1 to 5, with 1 being the lowest score and 5 the highest. Figure 11 shows the results of our user study. From Figure 11(a), we note that a majority of users assigned a score of 4, while the average score was 3.57. This value is above average, which indicates that our messenger is user-friendly. From Figure 11(b), we note that most users found our messenger to be better than 2D messengers. The average score of 4.05 indicates that users feel that the application is dynamic in communicating with other users. If (a) and (b) were surveys of the usability of simple 3D messengers, Figure 11(c) was a survey of whether existing 2D could be replaced. However, (c)'s evaluation did not seem to replace 2D messengers at the moment, as it received a lower rating than However, through Figure 11(d), our messenger has shown us the potential for future development as the next messenger. Rather than competitiveness in the present, where smartphones are at the center, our messenger in the future, especially on wearable devices, can certainly be seen as an advantage.
|
Reviewer 3 Report
First of all, I want to congratulate to authors for this manuscript. I would like to focus in some topics to improve the quality:
1) Table 1 is correct? Whatsapp does not emojis?
2) Section 2 is poor..
3) What are the strengths of this method? What is the main difference from Whatsapp for example?
4) User validation is poor described. It is necessary to increase to justify your proposal.
Author Response
1) Table 1 is correct? Whatsapp does not emojis?
Answer: We get the data from Wikipedia which has comparison table of various Chatting application such as whatsapp, line, facebook, etc. They says most of 2D chatting applications have no emojis. However, these days, while applications offer services similar to emojis, it is still closer to emoticons. In this sense, the reference[23] also states that whatsapp is not servicing emoji.
2) Section 2 is poor.
Answer: The research on 3D chat apps could not be found (usually only commercially developed), so other relevant parts were investigated and the reference was drawn up. As you said, there seemed to be a lot of deficiencies, so we reinforced a lot. The reinforced parts are as follows:
While research on 3D messengers has not been done much, there have been several studies of 3D avatars and 3D emotional expressions. A study by Ann J. et al has studied the visualizing changes in users' emotional states. The user's emotional information was inputted and printed out with the avatar's facial expression[26]. A similar study can be seen by Ronan B. et al, which also suggests how to efficiently map facial expressions to them based on the Arousal-Valence emotional model[27]. Both studies have conducted 3D representation of emotions, but because our research is an avatar expression on smart devices, it is different from these studies that are expressing avatar emotions on PCs. Also, Avatar's emotional expression is one of the advantages of 3D messenger, and our research has a big difference: the use of messenger on a wearable glass. Most of the abovementioned studies have used AR in conveying information only undirectionally. However, in our study, we develop a messenger that affords two-way AR-based communication. In our approach, emotional expressions are also added to provide a more dynamic interaction between users.
|
3) What are the strengths of this method? What is the main difference from Whatspp for example? (contribution)
Answer: The biggest difference from 2D messenger is interface. Currently, if it is a messenger using mobile phones, it is expected that future messengers will change with the development of wearable devices. Therefore, we can say that we focused on the development of next-generation messengers. We explained the contribution of our research and thought that the advantages of it were shown through validation. However, the validation seemed to be lacking, so the user-study was further reinforced and added. That part is added to the latter part of Chapter 5, and a description of the contribution is added to the lower part. That part is as follows.
Part of chap 1. In this context, we propose a new messenger that improves on the static presentation threshold of existing text-based messengers. The application offers a chat environment based on augmented reality (AR) technology with the use of wearable devices. Here, we note that wearable devices are used in virtual reality (VR) as well as AR technologies [1, 2, 3, 4]. AR can improve on the shortcomings of the 2D environment and afford the use of dynamic and nonverbal expressions. In this study, users used smartglasses, a wearable device, which facilitates the generation of an optimal user interface (UI) for an immersive experience[5, 6]. In addition, we applied an effective emotional expression method based on various emojis to utilize the appropriate augmentation method for the smartglasses by utilizing the Vuforia AR software development kit (SDK). Subsequently, we selected a dedicated server model for the network environment required for the messenger and implemented the network model using application programming interfaces (APIs) provided by Unity. As the current 2D messenger is used on mobile phones, the inconvenience is unknown. If everyone carries a mobile phone, the use of 2D messengers can continue to be used. But just as smartwatches use more smartwatches than ever before, it is certain that there will be an increase in the use of wearable devices. Wearable devices will be a 3D form of use interface, and we have conducted our research focusing on this part.
Part of chap 6. We proposed an AR-based emotional messenger to overcome the static conversation constraints of existing text-based messengers. The highlights of our study are as follows: First, our utilization of smartglasses allows users to experience AR and effectively immerse themselves in the chat environment. The use of messenger using smart glass is expected to increase in usability as IT devices develop in the future. It also provides a wider field of view compared to smartphones, so it can prevent problems that may arise from smartphones. For example, an accident occurs while walking without looking forward while using a messenger with a smartphone
|
4) User validation is poor described. It is necessary to increase to justify your proposal (Validation).
Answer: We have verified our results by adding the number of people in the existing user study and adding two other surveys. The added verification methods and contents are added to Figure 11 and are as follows.
We conducted a user study to compare the performance of our messenger service relative to that of 2D messengers. The users comprised 35 students who were in their 20s and were asked 4 questions each. These students were chosen because individuals in this age group form the dominant section of messenger users relative to other age groups. The four questions were: 1. Is the messenger convenient to communicate with other people? 2. Is the messenger dynamic when compared with the other messengers based on 2D text? 3. Would you replace the 2D messenger you are using now? 4. Does it seem appropriate for the next generation of messengers? Users could assign scores ranging from 1 to 5, with 1 being the lowest score and 5 the highest. Figure 11 shows the results of our user study. From Figure 11(a), we note that a majority of users assigned a score of 4, while the average score was 3.57. This value is above average, which indicates that our messenger is user-friendly. From Figure 11(b), we note that most users found our messenger to be better than 2D messengers. The average score of 4.05 indicates that users feel that the application is dynamic in communicating with other users. If (a) and (b) were surveys of the usability of simple 3D messengers, Figure 11(c) was a survey of whether existing 2D could be replaced. However, (c)'s evaluation did not seem to replace 2D messengers at the moment, as it received a lower rating than However, through Figure 11(d), our messenger has shown us the potential for future development as the next messenger. Rather than competitiveness in the present, where smartphones are at the center, our messenger in the future, especially on wearable devices, can certainly be seen as an advantage. |
Round 2
Reviewer 1 Report
My comments have not been taken into account. I still insist on limited engagement of AR technology and insufficient comparison with similar systems and applications.
Author Response
1) Limitation of engagement of AR technology
Answer: The reason for using Smart Glass is not simply to access AR-based messengers. Ar Messenger has the advantage of allowing users to express their emotions and become more immersed in content. The reason for using smart glass here is to move away from the commercialized platform to the next generation of wearable devices. Also, smart glass is representative of devices that can enjoy 3D that have ease of carrying. To sum up, the reason why AR messenger was produced using smart glass is because it was expected that smart glass would be the most commercialized device among 3D AR devices. We added this part to end of the introduction
If everyone carries a mobile phone, the use of 2D messengers can continue to be used. But just as smartwatches use more smartwatches than ever before, it is certain that there will be an increase in the use of wearable devices. Wearable devices will be a 3D form of use interface, and we have conducted our research focusing on this part. Smart glass is expected to be the most commercial among 3D AR-enabled devices, so in this study, AR messenger can be used through smart glass. |
2) Insufficient comparison with similar systems and applications.
Answer: There are several studies that can be compared when the emotional-based AR messenger developed by our smart glass is divided into parts. I mentioned it in a related study. Compared to each cipher, there are advantages of our studies. https://venturebeat.com/2019/04/07/remote-ar-will-make-it-so-we-can-work-or-play-anywhere/ shows results as an AR messenger, but it shows messengers through computers, not wearable devices. In other words, wearable devices do not give users the feeling of experiencing them. Comparing with Ahn et al's work and Boulic et al's study, it can be compared in terms of emotional messengers, similarly, there is the advantage of developing a wearable device. If you compare it with the messenger that exists in AR games, it is just an exchange of information for simple communication for the game, and communication is not as smooth as we are. So the parts shown in our results can be said to be different from each application, and these developments are mentioned in the related work.
While research on 3D messengers has not been done much, there have been several studies of 3D avatars and 3D emotional expressions. A study by Ann J. et al has studied the visualizing changes in users' emotional states. The user's emotional information was inputted and printed out with the avatar's facial expression[26]. A similar study can be seen by Ronan B. et al, which also suggests how to efficiently map facial expressions to them based on the Arousal-Valence emotional model[27]. Both studies have conducted 3D representation of emotions, but because our research is an avatar expression on smart devices, it is different from these studies that are expressing avatar emotions on PCs. Also, Avatar's emotional expression is one of the advantages of 3D messenger, and our research has a big difference: the use of messenger on a wearable glass. Most of the abovementioned studies have used AR in conveying information only undirectionally. However, in our study, we develop a messenger that affords two-way AR-based communication. In our approach, emotional expressions are also added to provide a more dynamic interaction between users.
|
Reviewer 2 Report
The authors have made some effort to address the reviewer comments, but their response is not comprehensive and does not address all of my comments. The authors seemed to have moved the work from the issue of emotional avatars to the use of an AR environment for chatting. Chatting in an AR environment is already known. The authors can consult:
https://spatial.io/
or
https://venturebeat.com/2019/04/07/remote-ar-will-make-it-so-we-can-work-or-play-anywhere/
for commercial examples of AR-based chat and interaction systems.
The experiment performed by the authors is designed to show that AR chat systems are superior to 2D systems. Given that there are commercial 3D AR chat and interaction systems currently available the authors still need to prove what is unique and different about their system. I would like to examine the author's claims to contribution:
"In this context, we propose a new messenger that improves on the static presentation threshold of existing text-based messengers. "
The authors must show something new here. https://spatial.io/ and https://venturebeat.com/2019/04/07/remote-ar-will-make-it-so-we-can-work-or-play-anywhere/ already describe similar systems. What is new about the proposed system? This must be clearly stated by comparison with these systems.
"The application offers a chat environment based on augmented reality (AR) technology with the use of wearable devices. Here, we note that wearable devices are used in virtual reality (VR) as well as AR technologies [1, 2, 3, 4]. "
This is not new, please see the links above.
"AR can improve on the shortcomings of the 2D environment and afford the use of dynamic and nonverbal expressions. "
The authors experiment attempts to show this. However, this has already been studied. For example,
https://www.sciencedirect.com/science/article/pii/S0148296318305319
What is unique about the author's approach to AR versus 2D? Is it the use of 3D glasses? This was done in https://venturebeat.com/2019/04/07/remote-ar-will-make-it-so-we-can-work-or-play-anywhere/ and https://spatial.io/. Are the authors claiming to be the first to study AR versus 2D for chat environments in a comprehensive way? If so, their experiment must be much more structured with set tasks for users to perform and controlled hardware. The users should be performing similar tasks in 2D environments as well as 3D environments. In order to test the overall hypothesis that AR environments versus 2D environments for chatting, the authors must perform a comprehensive experiment testing different forms of 2D chat versus different forms of 3D AR chat. Giving users one AR chat system to interact with in an uncontrolled setting and then simply asking them to answer some questions does not prove the hypothesis that AR chat is better than 2D chat.
"In this study, users used smartglasses, a wearable device, which facilitates the generation of an optimal user interface (UI) for an immersive experience[5, 6]. "
This has been done before. Why is the system presented by the authors better than other 3D chat systems such as
https://spatial.io/
or
https://venturebeat.com/2019/04/07/remote-ar-will-make-it-so-we-can-work-or-play-anywhere/
?
"In addition, we applied an effective emotional expression method based on various emojis to utilize the appropriate augmentation method for the smartglasses by utilizing the Vuforia AR software development kit (SDK)."
This has also been studied before:
Ahn, Junghyun & Gobron, Stéphane & Garcia, David & Silvestre, Quentin & Thalmann, Daniel & Boulic, Ronan. (2012). An NVC Emotional Model for Conversational Virtual Humans in a 3D Chatting Environment. 7378. 47-57. 10.1007/978-3-642-31567-1_5.
Why are the presented emotional avatars superior to previous work? Please explain clearly. No effort has been made to explain this in the revised text.
In summary:
If the authors unique contribution is an experiment showing that AR chat systems are superior to 2D chat systems in general then they should:
Compare multiple existing AR chat systems including their own against 2D chat systems in a series of controlled experiments involving users performing similar comparable experiments in a 2D environment vs a 3D environment and reporting on direct comparisons for the different tasks. An unstructured test environment with an AR system followed by a survey doesn't prove that all AR chat systems are superior to 2D systems.
If the authors wish to prove that their particular AR system is superior to other AR chat systems, then they should:
Compare against other AR chat systems, not simply 2D systems.
If the authors wish to prove that their emotional avatars produce a more effective AR chat experience, then they should:
Compare their system against other AR chat systems that do not include emotional avatars or against their own system without the emotional avatars. They also must frame their work clearly against other work on emotional avatars in 3D environments such as: Ahn, Junghyun & Gobron, Stéphane & Garcia, David & Silvestre, Quentin & Thalmann, Daniel & Boulic, Ronan. (2012). An NVC Emotional Model for Conversational Virtual Humans in a 3D Chatting Environment. 7378. 47-57. 10.1007/978-3-642-31567-1_5.
Author Response
1) If the authors unique contribution is an experiment showing that AR chat systems are superior to 2D chat systems in general then they should:
Compare multiple existing AR chat systems including their own against 2D chat systems in a series of controlled experiments involving users performing similar comparable experiments in a 2D environment vs a 3D environment and reporting on direct comparisons for the different tasks. An unstructured test environment with an AR system followed by a survey doesn't prove that all AR chat systems are superior to 2D systems.
Answer: We conducted various surveys to show that our AR chat system is superior to the 2D chat system, but we felt that it was not enough for what you said. However, it was not easy to test them in a similar environment in the first place because 2D and 3D messengers are systems of different environments. Therefore, we did a survey with the unstructured test environment you are talking about. In particular, the reason why we developed 3D AR messenger was because it had potential for future development. I've told you before.2D instant messaging is much more useful for people who currently have mobile phones. However, just as smartwatches become increasingly commercialized, it is expected that in the future, wearable devices as well as smart glass will be able to replace mobile phones. We developed 3D AR messenger in advance for this occasion. So the survey was forced to ask "Is it easy for users to use?" or "Is it a messenger for future use?" I will continue to think deeply about what you told me in future research. Thank you for telling me about this.
2) If the authors wish to prove that their particular AR system is superior to other AR chat systems, then they should:
Compare against other AR chat systems, not simply 2D systems.
Answer: Thank you for letting us know the various studies we can find. Comparing with Ahn et al's work and Boulic et al's study, it can be compared in terms of emotional messengers, similarly, there is the advantage of developing a wearable device. If you compare it with the messenger that exists in AR games, it is just an exchange of information for simple communication for the game, and communication is not as smooth as we are. So the parts shown in our results can be said to be different from each application, and these developments are mentioned in the related work.
While research on 3D messengers has not been done much, there have been several studies of 3D avatars and 3D emotional expressions. A study by Ann J. et al has studied the visualizing changes in users' emotional states. The user's emotional information was inputted and printed out with the avatar's facial expression[26]. A similar study can be seen by Ronan B. et al, which also suggests how to efficiently map facial expressions to them based on the Arousal-Valence emotional model[27]. Both studies have conducted 3D representation of emotions, but because our research is an avatar expression on smart devices, it is different from these studies that are expressing avatar emotions on PCs. Also, Avatar's emotional expression is one of the advantages of 3D messenger, and our research has a big difference: the use of messenger on a wearable glass. Most of the abovementioned studies have used AR in conveying information only undirectionally. However, in our study, we develop a messenger that affords two-way AR-based communication. In our approach, emotional expressions are also added to provide a more dynamic interaction between users.
|
3) If the authors wish to prove that their emotional avatars produce a more effective AR chat experience, then they should:
Compare their system against other AR chat systems that do not include emotional avatars or against their own system without the emotional avatars. They also must frame their work clearly against other work on emotional avatars in 3D environments such as: Ahn, Junghyun & Gobron, Stéphane & Garcia, David & Silvestre, Quentin & Thalmann, Daniel & Boulic, Ronan. (2012). An NVC Emotional Model for Conversational Virtual Humans in a 3D Chatting Environment. 7378. 47-57. 10.1007/978-3-642-31567-1_5.
Answer: Comparing to ahn's study that you told me, it was similar to the part where I was doing emotional ar chats. In other words, they can use their skills to match ours. But here we have the distinction of using wearable devices. Our goal was to make our research more commercialized in the future than the existing research. When I used the messenger without emotion and the system with emotion expression that you said to users, it was said that it was very boring to have no emotion expression on the 3D messenger. We don't think about comparing it to nothing, because we think one of the important points of 3D messenger is the expression of emotion like emoji. Also, since most users evaluated the need for emotional expression, they did not write it down directly in the paper. However, I think this survey provided the basis for the 3D messenger = emoji we thought. Thanks to you, I can get one better evidence.
Reviewer 3 Report
I agree with responses.
Author Response
Thanks for your helpful review comments
Round 3
Reviewer 1 Report
Authors have tried to take into account the suggestions proposed in the latter revision. However, the essence of the suggestions was supposed to trigger major changes in the contribution of the paper.
Regarding AR technology, the effort to present the application as AR-based is still misleading, as AR only is exploited during the initial phase of the application. A change should be made in the implementation of the application. Otherwise, the AR component as is should be ommitted as it dows not offer any useful functionality.
Regarding the comparison to existing systems, two other applications have been presented and compared. However, in the previous review more systems have been suggested and a more thorough, feature-based comparison among similar applications is needed in order to justify the innovation of the proposed application.
Author Response
1) The Usage of Smartglass in our research
Answer: In our study, the biggest difference between chatting using smart glass is that most of the existing messengers can be typed into buttons for simple communication. Additional necessary messengers can use the existing key button as well as send messages through voice recognition. We added it because it was the button method and voice input method that we used the most in carrying out AR messenger using smart glass.
4. Result … In this study, various ways of expressing opinions existing in existing messengers were expressed in synchronizing with augmented reality space. Smart glass has different input/output methods from existing devices, so it allows users to type characters through voice recognition. For keystrokes of traditional AR glass (see Figure 10(a)), there may be disadvantages of being uncomfortable or slow when users type. We did most of the methods with buttons, and in the case of chats requiring keystrokes, we added the ability to convert voice data into text through voice recognition. At this time, Google Cloud Speech"[28] was used, which had a high recognition rate among existing APIs. Like Figure 10(b), if we recognize the voice in Google Voice Recognition window, we can conveniently message the words we want to AR environment without typing. + Figure 10 is added & Figure 10,11 is move to Figure 11,12 |
2) Comparison with other application
Answer: There is no comparison study with AR message using AR glass, our study, but there are several comparable similar applications when each is broken down and compared. Similar applications may include emotional expression [26], avatar chatting[31], chatroom message [24,25], etc. In a large sense, it is called AR messenger, but our research is the most representative of the research that makes users feel like they are experiencing it in person. This is because the user uses the messenger using smart glass. As the research in [29,30] proved, smart glass helps users focus on messengers through hands-on experiences. Also mentioned in the first table with the existing 2D chatroom. Emoji's emotional expression is much freer and better expressive. Each was compared and referred to in Discussion. The changes are as follows.
Our research is the first to develop 3D AR Messenger using smart glass. I would like to compare our contributions with some similar studies. The biggest difference from the existing 2D messenger is that 3D chatting through avatars is possible. While today's messenger may be more efficient to chat over mobile phones, the time will come for 3D chatting to become more convenient as more wearable devices evolve [29,30]. Also, as 3D chatting is done, the theme of the chatting background, emoji that can express emotions in three dimensions, can be used. Compared to Avatar Chat [31], our research is done through smart glass. If a user enters a chat room directly through a smart glass, he or she can enter the avatar and chat more intimately, which can arouse interest. The research on emoji [26] is ultimately more likely to be included in our messengers and developed. [26] is different from our research because it is not conducted on smart devices, although it expresses emotion by reading the user's face. As a result, we developed 3D emotional messengers through wearable devices, especially smart glass that users can immerse in, which is a future-oriented messenger. |

Reviewer 2 Report
I have read the response. The work can now be published.
Author Response
Thanks for your comments